# Effects of Soybean Phosphate Transporter Gene *GmPHT2* on Pi Transport and Plant Growth under Limited Pi Supply Condition

**DOI:** 10.3390/ijms241311115

**Published:** 2023-07-05

**Authors:** Xiaoshuang Wei, Xiaotian Xu, Yu Fu, Xue Yang, Lei Wu, Ping Tian, Meiying Yang, Zhihai Wu

**Affiliations:** 1Faculty of Agronomy, Jilin Agricultural University, Changchun 130118, China; weixiaoshuang@jlau.edu.cn (X.W.); 20210001@mails.jlau.edu.cn (X.X.); tianping@jlau.edu.cn (P.T.); 2College of Life Sciences, Jilin Agricultural University, Changchun 130118, China; 20210020@mails.jlau.edu.cn (Y.F.); xueyang840316@163.com (X.Y.); lwu@jlau.edu.cn (L.W.); 3National Crop Variety Approval and Characterization Station, Jilin Agricultural University, Changchun 130118, China

**Keywords:** *GmPHT2*, soybean, phosphate transporter, low-phosphate stress, subcellular localization, expression profile

## Abstract

Phosphorus is an essential macronutrient for plant growth and development, but phosphate resources are limited and rapidly depleting due to massive global agricultural demand. This study identified two genes in the phosphate transporter 2 (PHT2) family of soybean by bioinformatics. The expression patterns of two genes by qRT-PCR at leaves and all were induced by low-phosphate stress. After low-phosphate stress, *GmPHT2;2* expression was significantly higher than *GmPHT2;1*, and the same trend was observed throughout the reproductive period. The result of heterologous expression of *GmPHT2* in Arabidopsis knockout mutants of *atpht2;1* shows that chloroplasts and whole-plant phosphorus content were significantly higher in plants complementation of *GmPHT2;2* than in plants complementation of *GmPHT2;1*. This suggests that *GmPHT2;2* may play a more important role in plant phosphorus metabolic homeostasis during low-phosphate stress than *GmPHT2;1*. In the yeast backfill assay, both genes were able to backfill the ability of the defective yeast to utilize phosphorus. *GmPHT2* expression was up-regulated by a low-temperature treatment at 4 °C, implying that *GmPHT2;1* may play a role in soybean response to low-temperature stress, in addition to being involved in phosphorus transport processes. *GmPHT2;1* and *GmPHT2;2* exhibit a cyclic pattern of circadian variation in response to light, with the same pattern of gene expression changes under red, blue, and white light conditions. *GmPHT2* protein was found in the chloroplast, according to subcellular localization analysis. We conclude that *GmPHT2* is a typical phosphate transporter gene that can improve plant acquisition efficiency.

## 1. Introduction

Phosphorus is a key component of cellular composition and is involved in many metabolic processes, making it one of the most important macronutrients for plant growth and development [1,2]. Inorganic phosphate (Pi) is the only form plant roots can assimilate. However, Pi in the soil is frequently suboptimal for plant growth due to cation interaction and microorganism conversion into organic forms [3,4,5]. Excessive fertilizer application has become common in agriculture to ensure crop productivity, resulting in many environmental issues [6]. Thus, there is an urgent need to breed plant varieties with high Pi acquisition and utilization capacity to reduce chemical fertilizers and improve agricultural sustainability.

Globally, many soils are deficient in Pi, and the concentration of Pi inside plant cells is typically 1000 times higher than outside [7,8]. To counteract the large concentration gradient at the root–soil interface, plants must have specialized transport systems that transport Pi from the soil to the cells [9]. The active phosphorus in the soil and the phosphorus used in various metabolic processes eventually require direct uptake and transport through the “phosphate transporter protein (PHT)”, a highly efficient plant root uptake and transport system that is responsible for the transport of inorganic phosphorus in plants under transcription factor regulation [10]. Many Pi transporter families are found in plants, including *PHT1*, *PHT2*, *PHT3*, *PHT4*, and *PHT5*, which are involved in Pi uptake by roots and remobilization in plants to adapt to low-phosphate conditions [2,11,12,13,14,15,16,17]. Most members of the *PHT1* family are responsible for the uptake and translocation of Pi [18]. Also, PHT1 family genes have been reported in wheat [19], barley [20], maize [21], and *Brassica napus* [22]. *OsPT2* is a low-affinity transporter gene associated with Pi translocation from roots to shoots [23]. *OsPT6*, *OsPT9*, and *OsPT10* are highly induced by Pi starvation and play roles in Pi acquisition and translocation [18,23].

As soybean (*Glycine max*) is a globally important and widely grown crop, phosphorus nutrition is important for accumulating dry matter in all soybean organs, particularly for increasing soybean seed yield [24,25]. Appropriate phosphorus concentrations in roots have been shown in studies to increase the root significantly and above-ground biomass of soybean, increasing root dry weight, root length, root surface area, and root volume [26], allowing for rapid growth and development of the soybean root system [27], providing a suitable foundation for mineral uptake and utilization by the soybean plant. Providing sufficient phosphorus nutrition to soybean during the reproductive period accelerates the photosynthetic phosphorylation process. It improves the photosynthetic capacity of the leaves, which can significantly increase the accumulation of dry matter in all soybean plant organs and the amount of dry matter transferred from the nutrient organs to the seeds, both of which are favorable for the formation of soybean seed yield [12]. Furthermore, the effective phosphorus concentration can control the fat-to-protein ratio in the seeds. Therefore, increasing soybean yield and quality in oilseed crops is only possible with phosphorus. Research on soybean phosphorus transport proteins has focused on the PHT1 family and the *PHO1* family, with the *PHT1* family being more comprehensively studied. Under low-phosphate stress, the expression of most of the *PHT1* family genes increases. Some soybean roots form a symbiosis with fungi, producing mycorrhizae to help soybean absorb phosphorus from the soil, which results from fungal infestation significantly increasing the expression of *GmPT8*, *GmPT9*, and *GmPT10* found that *GmPT5* is localized in the vascular system of soybean roots and root nodules and is a phosphorus high-affinity phosphorus transporter protein involved in the transport of phosphorus from soybean roots to root nodules and that *GmPT5* is a key gene in the dynamic balance of phosphorus in root nodules. There is a lack of systematic studies on other families of soybean [28,29]. 

Arabidopsis and rice each have one member of the *PHT2* gene family. *AtPHT2*;1 mediates Pi transport into chloroplasts [28]. *OsPHT2;1* aids Pi accumulation and translocation in rice, a putative low-affinity phosphate transporter gene [30]. Bioinformatics analysis was used to identify two soybean genes homologous to Arabidopsis *AtPHT2;1*, named *GmPHT2;1* and *GmPHT2;2*. They were cloned, and their biological functions were characterized to study their effects on soybean photosynthesis. This will lay the groundwork for understanding the regulatory mechanism of soybean *PHT2* genes in phosphorus uptake and transport and provide candidate genes for future use of genetic engineering to create high-yielding, high-quality soybean varieties.

## 2. Results

### 2.1. Phylogenetic Analysis of GmPHT2 Gene Family

*PHT2* sequences were identified and retrieved from 49 land plants (Appendix A), and the phylogenetic connections between *OsPHT2* family proteins and *Arabidopsis thaliana* PHT2 proteins were investigated. Based on genome database searches and phylogenetic analysis, The findings revealed two orthologs of *GmPHT2* in soybean, which we identified as *GmPHT2;1* and *GmPHT2;2*; they are clustered together (Figure 1). This suggests that the functions of *GmPHT2;1* and *GmPHT2;2* may be similar.

### 2.2. Expression Patterns of GmPHT2 Genes in Soybean

To identify the expression patterns of *GmPHT2* genes, we used gene microarray technology to examine the expression patterns of *GmPHT2* family genes during the growth stage. Both *GmPHT2;1* and *GmPHT2;2* genes were most highly expressed in the leaves of soybean, as shown in Figure 2, with *GmPHT2;2* gene expression in the leaves at 97.66, much higher than *GmPHT2;1* gene expression in the leaves at 56.46. This was followed by green pods, stem tips and flowers, and root and root hair regions, all of which were expressed at less than 5, significantly lower than the above tissues (Appendix A). This indicates that *GmPHT2;1* and *GmPHT2;2* have important roles in leaves and other green tissues.

### 2.3. Subcellular Localization of Gmpht2 Family Member

To determine the subcellular localization of the *GmPHT2s*, full-length complementary DNAs (cDNAs) of *GmPHT2* genes were fused to GFP and transiently expressed in *Arabidopsis thaliana* protoplasts. *GmPHT2;1* and *GmPHT2;2* fluorescent signals were detected in the chloroplast (Figure 3). These findings indicated that *GmPHT2s* were localized in the chloroplast.

### 2.4. Pi Transport Activity of GmPHT2 Genes in Yeast

All two members of the soybean *GmPHT2* family can compensate for the growth deficiency of the PAM2 strain, a yeast Pi transport mutant deficient in the high-affinity Pi transporter genes *PHO84* and *PHO89*, and facilitate Pi transport across membranes in yeast [15]. Thus, we hypothesized that the *GmPHT2* proteins found in soybean might also have Pi transport functions. To test this hypothesis, we expressed *GmPHT2;1* and *GmPHT2;2* driven by the ADH1 promoter in a yeast mutant strain PAM2, empty vector, and wild-type yeast were applied to YNB solid medium with a phosphorus concentration of 20 mM as a control. The findings are depicted in Figure 4a. The transfer of the target segments *GmPHT2;1* and *GmPHT2;2* at a gradient of 10^2^ inoculated yeast demonstrated that *GmPHT2;1* and *GmPHT2;2* could partially replace the yeast mutant PAM2 phosphorus transport.

To assess the ability of phosphorus transport capacity of GmPHT2;1 and GmPHT2;2, yeast expression vectors pYES2-GmPHT2;1 and pYES2-GmPHT2;2 for the GmPHT2;1 and GmPHT2;2 genes were constructed (Appendix A). It showed that the pYES2 cells with an empty vector did not grow. Yeast transformed with pYES2-GmPHT2;1 and pYES2-GmPHT2;2 grew better on all plates than those transformed empty vectors but with slightly fewer colonies than wild-type yeast. However, compared to 10^2^/plate and 10^4^/plate, the results for 10^3^/plate were more pronounced. (Figure 4a). Preliminary indications are that GmPHT2;1 and GmPHT2;2 have the phosphorus transport capacity of the complementary yeast inorganic phosphorus transport mutant PAM2.

Yeast Pi uptake defective mutant strain PAM2 (Δpho84Δpho89) was cultured in liquid YNB medium at low phosphorus concentration (10 μM) with Gal (inducer of pYES2) and Glu as a carbon source for yeast transformed with pYES2, pYES2-GmPHT2;1, and pYES2-GmPHT2;2, respectively, at OD600 The results of color development for four gradients of 10^−1^, 10^−2^, 10^−3^, and 10^−4^ dilutions of the bacterial solution at 0.8 are shown in Figure 4b. The color of the liquid medium with Glu as the carbon source was always reddish brown and did not change. The color of the liquid medium with Gal as the carbon source gradually became darker from 10^−1^ to 10^−4^, and the yeast cultures transformed with pYES2-GmPHT2;1 and pYES2-GmPHT2;2 were orange-yellow in color after 10-fold dilution. This indicates that GmPHT2;1 and GmPHT2;2 have the phosphorus transport capacity of the complementary yeast inorganic phosphorus transport mutant PAM2. This is consistent with the results for yeast solid media.

### 2.5. Expression Patterns of GmPHT2 Genes in Response to Different Treatments

We also looked into whether Pi input affected *GmPHT2* gene transcription. Because of MYB transcription factors, phosphate starvation response can bind to PHR1-binding sequences (P1BS) to activate the transcription of Pi starvation-induced genes, and the P1BS is present in the promoter of most Pi starvation-induced genes [31]. We first examined the P1BS in *PHT2* gene promoters using PLACE (http://www.dna.affrc.go.jp/PLACE/, accessed on 9 May 2020). It was discovered that the promoter regions of *GmPHT2;1* and *GmPHT2;2* contain Box4, ABRE, ARE, and CGTCA motif in addition to the usual amount of low-phosphate stress-related elements P1BS (Figure 5a). It suggested that the expression of most *GmPHT2s* could be influenced by Pi availability. Consequently, we investigated the expression pattern of *GmPHT2* in soybean at the meristem, pod bulge, and maturity stages with persistently low and persistently normal phosphorus levels, as shown in Figure 5b. *GmPHT2;1* and *GmPHT2;2* gene expression did not vary significantly between soybean periods treated with continuous normal phosphorus. At the branching and podding phases, the expression of the *GmPHT2;2* gene was significantly higher than that of the *GmPHT2;1* gene in soybean treated with continuous low phosphorus. This suggests that the expression of both *GmPHT2;1* and *GmPHT2;2* genes is increased in response to sustained low phosphorus levels in soybean, with the *GmPHT2;2* genes possibly playing a more significant role.

We discovered the low-temperature response element LTR in the promoter region of the *GmPHT2;1* gene during the promoter analysis, so the soybean seedlings were treated with the low temperature at 4 °C for 12 h at the seedling stage, and the results are shown in Figure 5c. *GmPHT2;1* and *GmPHT2;2* gene expression increased with treatment duration, peaking at 6 h, and then decreasing. *GmPHT2;1* gene expression was considerably higher than *GmPHT2;2* genes during the low-temperature treatment.

The promoter elements of *GmPHT2;1* were more abundant than those of *GmPHT2;2*, with 10 and 7 light-responsive elements, respectively. This could imply that *GmPHT2;1* and *GmPHT2;2* react to light and low-phosphate stress processes. Figure 5e–g depict gene expression under white, red, and blue light after 12 h of darkness and return to the light. Within 3 h, the expression of the *GmPHT2;1* gene was higher than that of the *GmPHT2;2* genes, as shown in the graph. After 6 h, the expression of the *GmPHT2;2* gene was gradually higher than that of the GmPHT2;1 gene. After 12 h, the expression of the *GmPHT2;2* gene was higher than that of the *GmPHT2;1* gene. The *GmPHT2;1* and *GmPHT2;2* genes exhibit the same expression pattern under different light conditions.

### 2.6. Heterologous Expression of GmPHT2 Enhanced Pi Uptake in Arabidopsis

Figure 6a–d demonstrate the plant phenotypes of identified Arabidopsis T2 generation seeds cultured for 60 days. The wild variety’s above-ground fresh and dry weights were significantly higher than the other plants. Still, the below-ground dry and fresh weights of the complementation and overexpressed plants were not significantly different. (Figure 6e,f). The primary root length was the same except for the mutant plants, which were not significantly different. (Figure 6g). Complementation and overexpressed plants had considerably larger rosette diameters than mutant plants but were smaller than wild-type plants (Figure 6h).

Five plants from each Arabidopsis species were chosen, and their phosphorus content was determined using the molybdenum blue colorimetric technique. Complementation and overexpressed plants had considerably lower phosphorus content than wild-type plants but significantly higher than mutant plants (Figure 7a). The phosphorus content of the chloroplasts and above-ground and below-ground parts of the plants in the back-complemented GmPHT2;2 plants were considerably higher than in the back-complemented GmPHT2;1 plant (Figure 7a,b). Measurements of photosynthetic indicators in Arabidopsis thaliana show that wild-type plants produced the highest net photosynthetic rate, while the knockout mutant produced the least. Plants with complementation GmPHT2;1 and GmPHT2;2 genes had greater net photosynthetic rates than mutants but significantly differed from the wild type. (Figure 7c). The differences in stomatal conductance, intercellular CO_2_ concentration, and transpiration rate between Arabidopsis species were insignificant (Figure 7d–f). It is suggested that the GmPHT2;1 and GmPHT2;2 genes restore the photosynthetic capacity of Arabidopsis mutant plants to some extent by increasing the phosphorus content of chloroplasts in the Arabidopsis mutant.

## 3. Discussion

Phosphate transporters are needed in plants for phosphate transport and uptake. This study systematically analyzed the entire *PHT2* gene family in soybean, which will allow researchers to identify candidate genes involved in phosphate use efficiency and abiotic stress response. Two *PHT2* genes have been identified in soybean, and they share significant similarities with Arabidopsis *PHT2* gene family members. This suggests that the *PHT2* genes in soybean and Arabidopsis are evolutionarily conserved and functionally similar. Since Daram et al. discovered the presence of the phosphorus transporter protein *PHT2* family, *AtPHT2;1* gene, in the model plant *Arabidopsis thaliana*, the biological function of the *PHT2* family in plants has garnered increased attention [32]. The *PHT2* family was found to have 12 similar transmembrane regions that encode amino acids with a large hydrophilic loop between the 8th and 9th transmembrane regions, a particularly long hydrophilic group at the N-terminus, and both belong to the PHO4 superfamily. A total of two *PHT2* family members were identified from soybean, and both *GmPHT2;1* and *GmPHT2;2* had conventional PHO4 superfamily structures when their conserved structural domains were examined (Appendix A).

Photosynthesis, the essential chemical reaction in living systems, occurs in chloroplasts [33]. Pi is an important substrate for photophosphorylation in chloroplasts, which converts ADP to ATP and thus promotes CO_2_ assimilation for phosphotriose synthesis. A series of phosphorus transport proteins/transporters found in the chloroplast membrane is important in maintaining chloroplast Pi homeostasis [34]. The *Arabidopsis thaliana AtPHT2;1* gene is primarily expressed in green plant tissues, with the greatest expression in plant leaves, implying that *AtPHT2;1* is involved in phosphorus transport within leaves [32]. The mutant *atPHT2;1* was discovered to be localized in the chloroplast, to have substantially lower phosphorus content in the chloroplast than the wild type, and to exhibit significant changes in plant growth, demonstrating that *AtPHT2;1* serves an important part in phosphorus translocation to the chloroplast. Huang et al. and Zheng et al. similarly studied the *PHT2* family *TaPHT2;1* and *OsPHT2;1* in wheat and rice and found that both genes were also localized to the chloroplast membrane. The present experimental results showed that *GmPHT2;1* and *GmPHT2;2* were localized to chloroplasts (Figure 3) [35,36].

Arabidopsis *AtPHT2;1* has high homology to the yeast phosphorus transporter protein gene *PHO84* at the nucleotide and amino acid level. Experiments using yeast mutants, *AtPHT2;1*, *TaPHT2;1*, and *OsPHT2;1*, restored phosphorus uptake by mutant yeast. Exogenously expressed *AtPHT2;1*, *TaPHT2;1*, and *OsPHT2;1* genes restarted phosphorus uptake by the yeast mutant on medium yeast plates at a phosphorus concentration of 10 mM. The inability to backfill the yeast mutant strain on yeast media with phosphorus concentrations less than 0.1 mM and 1 mM demonstrates that *AtPHT2;1*, *TaPHT2;1*, and *OsPHT2;1* are low-affinity phosphorus transport proteins [34,35,36,37,38]. The functional back-complementation assay of yeast mutants in this study further demonstrated that both proteins, *GmPHT2;1* and *GmPHT2;2*, can transport phosphorus (Figure 4). 

In this research, gene expression analysis of soybean leaves at various periods showed that the expressions of *GmPHT2;1* and *GmPHT2;2* were significantly higher in soybean at different fertility stages after low phosphorus treatment. *GmPHT2;1* and *GmPHT2;2* expression was higher at the meristem and embryo stages than at other times. Fischer and Huang et al. demonstrated that both *AtPHT2;1* and *TaPHT2;1* have a circadian expression pattern, with daytime expression increasing and nighttime expression decreasing [34,35]. In the present study, promoter analysis showed that multiple light-responsive elements were present in the promoter regions of both *GmPHT2;1* and *GmPHT2;2* genes. When *GmPHT2;1* and *GmPHT2;2* were subjected to continuous 72 h diurnal treatment and gene expression was measured every 12 h, they exhibited the same circadian pattern (Figure 5). Huang et al. discovered that *OsPHT2;1* is implicated in the metabolic pathway that regulates light damage in rice plants [35]. This research induced soybean by different light qualities, and the expression of *GmPHT2;1* and *GmPHT2;2* exhibited different patterns. The expression of *GmPHT2;1* was higher in the first 6 h, while the expression of *GmPHT2;2* regained higher than that of *GmPHT2;1* with time, the reaction rate to different light was significantly *GmPHT2;1*. Furthermore, low-temperature induction studies on soybeans revealed that *GmPHT2;1* and *GmPHT2;2* responded to low temperatures. This implies that while *GmPHT2;1* and *GmPHT2;2* can translocate phosphorus, *GmPHT2;1* responds before *GmPHT2;2* when the environment changes to ensure the plant’s phosphorus requirements.

In the model plant, *Arabidopsis thaliana*, the phosphorus content of chloroplast cells in *atpht2;1* mutant plant is significantly lower than in the wild type [32]. Back-complementation of the *GmPHT2;1* and *GmPHT2;2* genes increased the phosphorus content of chloroplasts in the leaves and the phosphorus content of the entire plant, particularly in the above-ground parts compared to the below-ground parts. These findings indicate that *GmPHT2;1* and *GmPHT2;2* are chloroplast-localized phosphorus transporter proteins essential in maintaining phosphorus homeostasis. Ayadi et al. demonstrated that the mutant *ospht2;1* rice plants grew considerably slower than the wild type [37]. In their study of *TaPHT2*, Secco and Poirier came to the same result that *TaPHT2;1* expression in wheat seedlings directly affected seedling development [38]. In this study, we discovered that Arabidopsis plants with defective phosphorus transport and those with back-complemented GmPHT2;1 and GmPHT2;2 genes were weaker than wild plants, with significant differences in plant rosette diameter. *atpht2;1* Arabidopsis leaves that were significantly smaller in size and petiole length than back-complemented plants (Figure 6a–d), this suggests that both genes can backfill the defect in atpht2;1. The same trend was observed for the plants’ total phosphorus content, with the plants’ above- and below-ground parts significantly higher than those of the deficient plants. The phosphorus content of Arabidopsis chloroplasts complementation of GmPHT2;1 and GmPHT2;2 genes were considerably higher, but it still differed from the wild type. The plants’ net photosynthetic rate recovered after complementation of the GmPHT2;1 and GmPHT2;2 genes, and Nakamori et al. discovered that phosphorus deficiency affected the photosynthetic electron transport chain, resulting in a significant reduction in photosynthesis and dry matter accumulation [39]. The *GmPHT2;1* and *GmPHT2;2* genes restored some of the photosynthetic capability of Arabidopsis mutant plants by increasing the phosphorus concentration of Arabidopsis mutant chloroplasts.

Most higher plants, such as rice [34], Arabidopsis [35,36], wheat [37,38,40,41], and tomato [42], only had one PHT2 family member, while bioinformatics analysis revealed the presence of two. The expression of *GmPHT2;2* was nearly twice as high as that of *GmPHT2;1* in soybean at the seedling stage, according to gene microarray data, and the findings verified that there were some differences between the two genes. *GmPHT2;2* expression was higher in soybean than *GmPHT2;1*, with smaller differences in expression at the branching and podding stages under normal phosphorus conditions and a clear tendency toward increased differences after low-phosphate stress.

The promoter region of the *GmPHT2;1* gene had one low temperature (LTR), two hormone classes (ABRE, CGTCA motif), one anaerobic (ARE), and a nine-light response element, according to promoter analysis. *GmPHT2;1* expression increased considerably after low-temperature treatment, and the expression of *GmPHT2;2* was less affected by low temperature. Although the promoter of *GmPHT2;2* does not contain an LTR, it is still induced at low temperatures; *GmPHT2;2* was very close to *GmPHT2;1*. We speculate that there may be similarities in the partial function of the two genes. Arabidopsis transformation studies revealed that plants back-complemented and overexpressing the *GmPHT2;2* genes outperformed those back-complemented and overexpressing the *GmPHT2;1* gene in terms of growth, phosphorus content, and net photosynthetic rate. We hypothesize that the two *PHT2* members in soybean are not redundant and that their functions have diverged during evolution, with *GmPHT2;1* functioning not only in soybean in response to low phosphorus but also in other biological processes such as low-temperature stress, whereas *GmPHT2;2* functions primarily in phosphorus transport. Meanwhile, GmPHT2s genes could have played a role in abiotic stress reactions. However, further research into the physiological roles of this family in soybeans is required.

## 4. Materials and Methods

### 4.1. Plant Materials and Growth Conditions

Laboratory-preserved soybean variety Chang Nong 26 (Validation No.: JI Audited Bean 2010004) was set at two phosphorus levels: normal phosphorus (0.5 mM) and low phosphorus (0.01 mM). The sand culture was used until the three-leaf stage of the soybean when it was transferred to hydroponic treatment. The culture was hydroponically grown for 12 h in 1/4 Hoagland nutrient solution. All the plants were grown in a greenhouse with a 12 h day (30 °C)/12 h night (22 °C) photoperiod, nearly 200 μmol/(m^2^·s) photon density, and humidity of about 60%. The normal phosphorus-treated soybeans continued to be divided into three treatments, which were: (1) Under normal phosphorus (0.5 mM) concentration, soybean after the three-leaf stage, a leaf from the bottom of three well-grown plants was taken at 12 h and 24 h daily for three consecutive days and stored at −80 °C. (2) Under the same culture conditions, the plants were divided into thirds and transferred to three light incubators for one day before exposure to continuous white, red, and blue light in the morning, shortly after dark. Three healthy plants were chosen at 0, 3, 6, 9, 12, and 24 h, and the bottom leaves were taken and stored at −80 °C. (3) The bottom leaves of suitably growing plants were stored at −80 °C for 24 days in a refrigerator at 0, 3, 6, 9, and 12 h after being transferred from the artificial climate chamber at the end of darkness under normal phosphorus (0.5 mM). 

Also, under greenhouse cultivation conditions, three suitable plants were selected at normal (0.5 mM) and low (0.01 mM) phosphorus concentrations at the branching, podding, and maturity stages, and their top leaves were stored at −80 °C.

### 4.2. Bioinformatic and Phylogenetic Analysis of GmPHT2 Gene Family

The protein sequences of PHT2s from various plant species were obtained from PLAZA (http://bioinformatics.psb.ugent.be/plaza/, accessed on 10 May 2020) and aligned using the Generous software. Phylogenetic analysis was conducted with CIPRES (www.phylo.org, accessed on 10 May 2020) using maximum-likelihood phylogenetic analysis with 1000 bootstrap replicates.

### 4.3. RNA Isolation, Quantification, and First-Strand cDNA Synthesis

The TAKARA Trizol Kit was used to extract total RNA. Total RNA was extracted from the leaves. A Nanodrop 2000 spectrophotometer was used to determine the quality and quantity of RNA. According to the manufacturer’s protocol, the first-strand cDNA was synthesized from 2 μg of total RNA using the PrimeScriptTMRT reagent Kit with gDNA Eraser (Takara, Japan) in a 20 μL reaction volume, DNA enzyme treatment is required to digest off gDNA before reverse transcription. It was then stored at −20 °C until further use.

### 4.4. Quantitative Real-Time PCR (qRT-PCR)

qRT-PCR was performed in 384-well plates on an ABI stepone plus machine (Roche, Basel, Switzerland). The 25 μL reaction mixture contained 1 μL cDNA and 2 μL each gene-specific primer (Appendix A) and 10 μL master mix (SYBR Green PCR Mastermix, Takara, Japan), water blank control. The cycling conditions were as follows: pre-denaturation at 95 °C for 10 min; then 40 cycles at 95 °C for 30 s (denaturation), 60 °C for 10 s (annealing), and 72 °C for 20 s (extension), followed by a melting curve analysis to verify the correct amplification of target gene fragments and the lack of primer dimers. The rice actin gene was used as a reference, and the transcript levels of genes were calculated according to the 2-ΔΔCt method [43]. Three biological replicates and three technical replicates were used in all qRT-PCR experiments.

### 4.5. Subcellular Localization of Genes

For subcellular localization analysis, the open reading frames (ORFs) of GmPHT2;1 and GmPHT2;2 were amplified and introduced into a p1302SNB vector containing 35S promoter and green fluorescent protein (GFP) fragments. The GFP was fused to the C-terminals of these genes (Appendix A). The GFP fusion plasmids were transformed into protoplasts by the polyethylene glycol (PEG)-mediated transformation. Protoplast preparation and transfection followed previously described procedures [44,45]. Confocal microscopy images were taken under a laser scanning confocal microscope (Leica TCS SP5, Leica, Wetzlar, Germany). Excitation and emission wavelengths Ex488/Em500-550, Ex516/Em600-650, and Ex587/Em602-652 were used for GFP, autofluorescence of chlorophyll, and red fluorescent protein (RFP), respectively.

### 4.6. Complementation of Yeast Pi Transport Mutant by PHT2 Genes

The coding regions of *GmPHT2;1* and *GmPHT2;2* were introduced into the BamHI and Xho I sites on a pYES2 vector to generate vectors for yeast complementation (Appendix A). These constructs and the empty vector were transformed into a yeast mutant strain PAM2, which is defective in Pi transport. Transformants were selected on a uracil-deficient medium. One colony was selected from each transformation strain and grown in the liquid SC-uracil medium. The mid-exponential growth phase cells were harvested and washed thrice with sterile water and resuspended to OD600 = 1. Equal volumes of 10-fold serial dilutions were spotted on yeast nitrogen-based (SM) medium of different Pi concentrations. Plates were incubated at 30 °C for 3 days. The composition of the solid medium is essentially the same as that of the SM medium, with the addition of galactose or glucose as the carbon source in the induction or non-induction medium, respectively, and the phosphorus content reduced from 6 mmol/L to 10 μmol/L. After 3 days of incubation at 30 °C, the ability of the GmPHT2;1, GmPHT2;2 genes to functionally complement the phosphorus uptake defective mutant yeast was determined by testing the growth of the recombinant yeast strain with bromocresol violet as a pH indicator.

### 4.7. Analysis of Promoters of GmPHT2 Family Genes

The gene sequence was acquired from the soybean genome database phytozome (V12.1), and the promoter region was a 2000 bp sequence upstream of the gene. The promoter element prediction website PlantCare (http://bioinformatics.psb.ugent.be/webtools/plantcare/html/, accessed on 20 May 2020) was used to predict promoter elements for all GmPHT members. Other than the fundamental promoter elements (TATA box and CAAT box), the promoter elements were mapped using the GSDS online mapping tool for promoter elements.

### 4.8. Vector Construction and Gene Transformation

To order *atpht2;1* Arabidopsis seeds, sequence alignment was performed using the TAIR website to identify the most similar gene to *GmPHT2;1* and *GmPHT2;2*, *AtPHT2;1*. The mutant seeds were planted in nutrient soil with wild-type seeds as a control and left to grow for about two months before being tested for pure mutant strains with assay primers LP: TTGTGAGGCCCAACCAAAAGAG; BP: GTCATCATCAATATCATCACCA; RP: AGGTGCTCTTTTGATGGGAACTCACGT, and seeds were harvested at maturity. By floral dipping, the backfill vectors pCAMBIA3301-GmPHT2;1 and pCAMBIA3301-*GmPHT2;2* were used to transform mutants and wild-type *Arabidopsis thaliana* [46]. T1 generation seeds were planted, PCR assays were performed 60 days later, plants that tested positive were harvested for seed, and T2 generation plants were used for subsequent experiments.

### 4.9. Phenotypic Identification of Arabidopsis Plants

*Arabidopsis thaliana* was regularly grown for 60 days. Three plants from each Arabidopsis species were chosen for the study. Chloroplast isolation was performed using the experimental technique of the Minute (TM) Chloroplast Isolation Kit from Sibel Biologicals, Inc. Phosphate concentration with the ascorbate-molybdate-antimony method [47]. The photosynthetic parameters of Arabidopsis: net photosynthetic rate, stomatal conductance, intercellular CO_2_ concentration, and transpiration rate, were measured using the LI-6800 technique of LI-COR, Lincoln, NE, USA.

### 4.10. Statistical Analysis

The experiments were repeated three times. The data were analyzed using SPSS Statistical 20.0 software, and the differences between groups were determined using Duncan’s test. Significant differences were represented by * for *p* ≤ 0.05.

## 5. Conclusions

In this study, we obtain two soybean phosphorus transporter protein PHT2 genes, GmPHT2;1, GmPHT2;2 by bioinformatics. RT-qPCR analysis showed that it was mostly expressed in the leaves, and all were induced by low-phosphate stress, and cis elements in these two transporters were found to be associated with light, low-phosphate stress, and circadian control. The expression patterns of GmPHT2 were characterized successfully in various treatments. Heterologous expression of GmPHT2 in Arabidopsis thaliana and results of yeast backfill assays show that GmPHT2 plays an important role in the balance of plant phosphorus metabolism that can improve plant acquisition efficiency. 

## Figures and Tables

**Figure 1 ijms-24-11115-f001:**
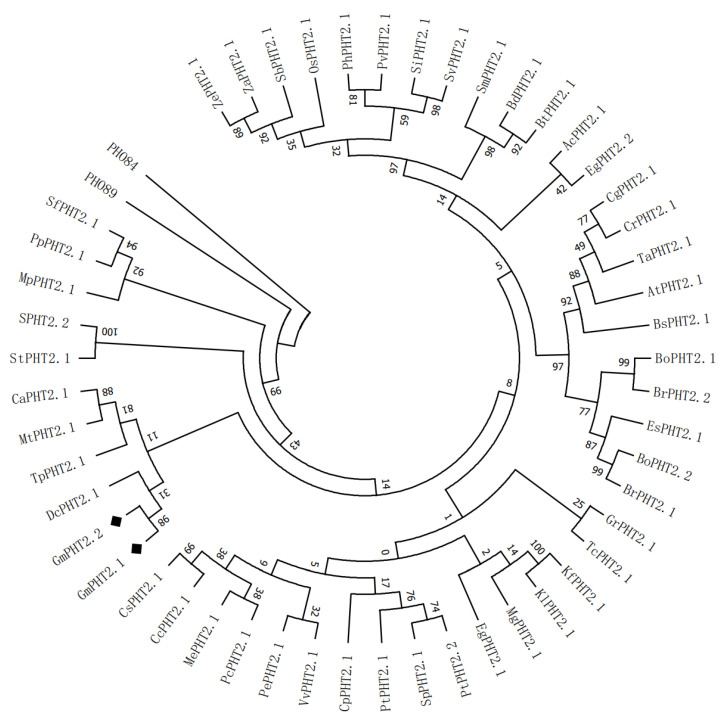
Phylogenetic tree of 52 *PHT2* family proteins from 49 species. ◆*GmPHT2;1* (Glyma.08G282100) and *GmPHT2;2* (Glyma.18G144100). Note: ZePHT2.1—*Zea mays* Ensembl-18; ZaPHT2.1—*Zea mays* PH207; SbPHT2.1—*Sorghum bicolor*; OsPHT2.1—*Oryza sativa*; PhPHT2.1—*Panicum hallii*; PvPHT2.1—*Panicum virgatum*; SiPHT2.1—*Setaria italica*; SvPHT2.1—*Setaria viridis*; SmPHT2.1—*Solanum melongena* L.; BdPHT2.1—*Triticum aestivum* L.; BtPHT2.1—*Brachypodium distachyon*; AcPHT2.1—*Aquilegia coerulea* EgPHT2.2—*Eucalyptus grandis*; CgPHT2.1—*Capsella grandiflora*; CrPHT2.1—*Capsella rubella*; TaPHT2.1—L.; AtPHT2.1—Arabidopsis thaliana Columbia; BsPHT2.1—Boechera stricta; BoPHT2.1—*Triticum aestivum Brassica oleracea capitata*; BrPHT2.2—*Brassica rapa* FPsc; EsPHT2.1—*Eutrema salsugineum*; BoPHT2.2—*Brassica oleracea capitata*; BrPHT2.1—*Brassica rapa* FPsc; GrPHT2.1—*Gossypium raimondii*; TcPHT2.1—; KfPHT2.1—*Kalanchoe fedtschenkoi*; KlPHT2.1—*Kalanchoe laxiflora*; *Theobroma cacao*; MgPHT2.1—*Mimulus guttatus*; EgPHT2.1—*Eucalyptus grandis*; SfPHT2.1—*Sphagnum fallax*; PpPHT2.1—*Prunus persica*; MpPHT2.1—*Marchantia polymorpha*; SpPHT2.2—*Salix purpurea*; StPHT2.1—*Solanum tuberosum*; CaPHT2.1—*Carica papaya*; MtPHT2.1—*Medicago truncatula*; TpPHT2.1—*Trifolium pratense*; DcPHT2.1—*Daucus carota*; GmPHT2.1—*Glycine max*; GmPHT2.2—*Glycine max* CsPHT2.1—*Citrus sinensis*; CcPHT2.1—*Citrus clementina*; MePHT2.1—*Manihot esculenta*; PcPHT2.1—*Ricinus communis*; PePHT2.1—*Prunus persica*; VvPHT2.1—*Vitis vinifera* Genoscope.12X; CpPHT2.1—*Carica papaya*; PtPHT2.1—*Populus trichocarpa*; SpPHT2.1—*Salix purpurea*; PtPHT2.2—*Populus trichocarpa*.

**Figure 2 ijms-24-11115-f002:**
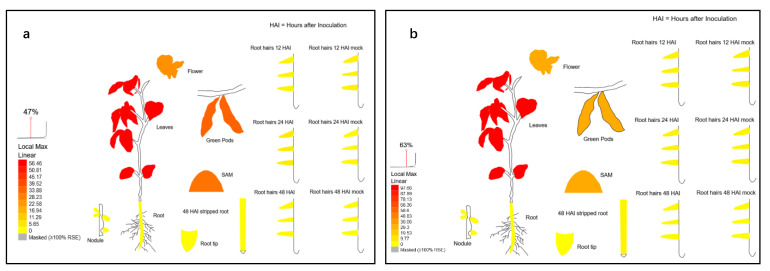
The gene chips expression analysis of *GmPHT2;1* and *GmPHT2;2*. (**a**), *GmPHT2;1* chip expression; (**b**), *GmPHT2;2*; (**a**), *GmPHT2;1* chip expression.

**Figure 3 ijms-24-11115-f003:**
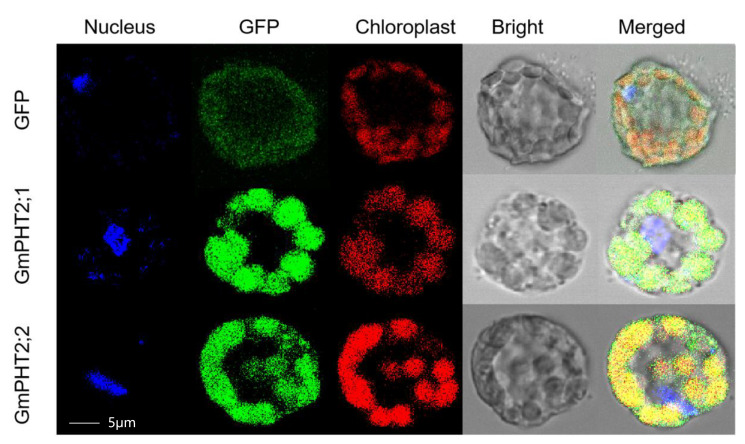
Subcellular localization of *GmPHT2;1* and *GmPHT2;2*. Transient expression of fusion protein *GmPHT2;1*-GFP, *GmPHT2;2*-GFP, and GFP in Arabidopsis mesophyll protoplasts was analyzed by confocal laser scanning microscope. GFP, dark field; Bright, under light; Chloroplast, red fluorescence indicates chloroplast autofluorescence; Merge, together with corresponding merged images. Scale bars: 5 µm.

**Figure 4 ijms-24-11115-f004:**
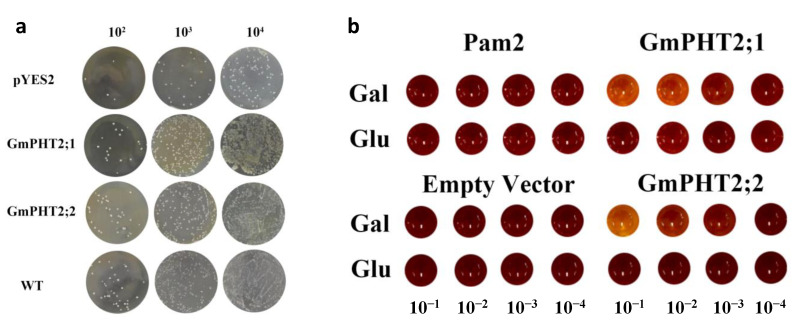
*GmPHT2* genes can confer Pi transport activity in yeast. (**a**) Complementary effects of *GmPHT2* family genes on yeast inorganic phosphate transport mutants; (**b**) the growth test of *GmPHT2* family gene transformed yeast PAM2 under liquid culture conditions. Gal, galactose, and Glu, glucose.

**Figure 5 ijms-24-11115-f005:**
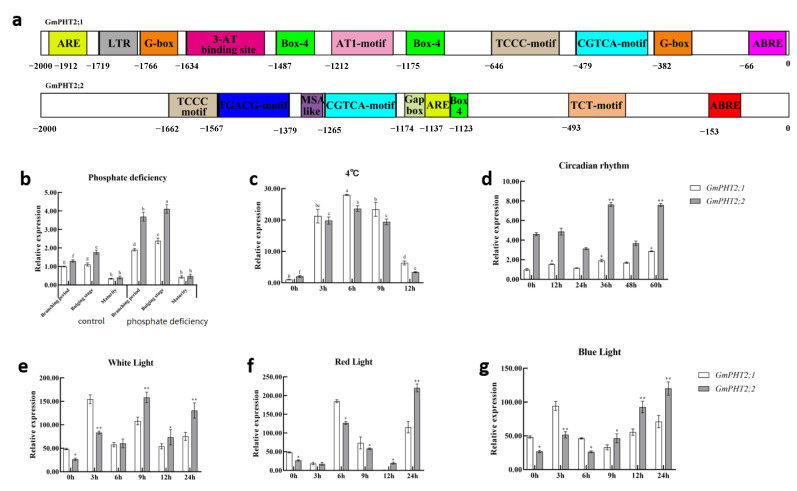
Expression patterns of *GmPHT2* genes in response to different treatments. (**a**) Promoter element analysis of *GmPHT2;1* and *GmPHT2;2*; (**b**) expression of *GmPHT2;1* gene, *GmPHT2;2* gene under low phosphorus conditions; (**c**) expression of *GmPHT2;1* and *GmPHT2;2* under 4 °C low-temperature treatment; (**d**) the rhythm expression of genes *GmPHT2;1*, *GmPHT2;2*; (**e**) the expression of *GmPHT2;1* and *GmPHT2;2* under white light; (**f**) the expression of *GmPHT2;1* and *GmPHT2;2* under red light; (**g**) the expression of *GmPHT2;1* and *GmPHT2;2* under blue light. * *p* < 0.05, ** *p* < 0.01. Different lowercase letters indicate significant differences at *p* < 0.05.

**Figure 6 ijms-24-11115-f006:**
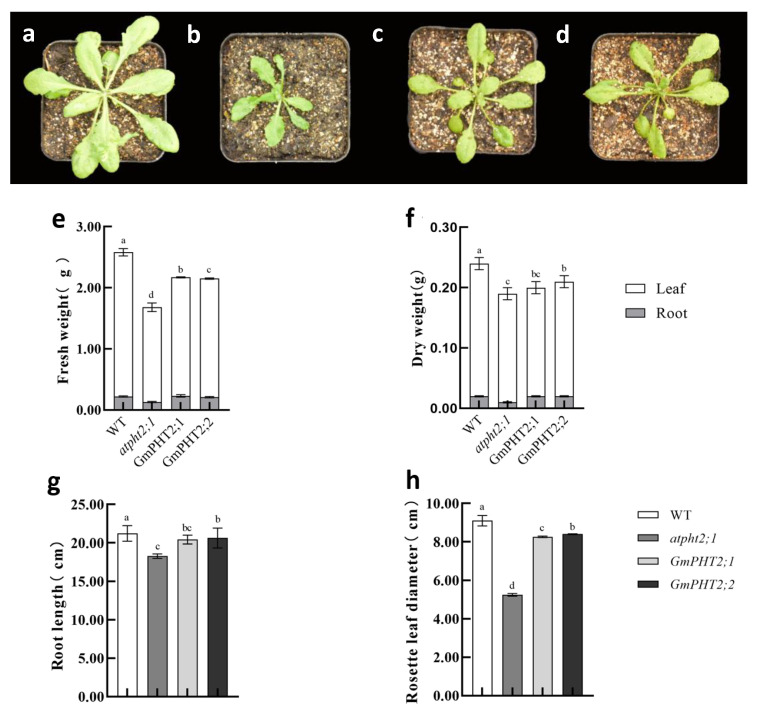
Phenotypic differences in different *Arabidopsis thaliana* and differences in physiological indicators. (**a**): Wild type; (**b**): *atpht2;1*; (**c**): complementation of *GmPHT2;1*; (**d**): complementation of *GmPHT2;2*; (**e**): fresh weight of different *Arabidopsis thaliana*; (**f**): dry weight of different *Arabidopsis thaliana*; (**g**): root length of different *Arabidopsis thaliana*; (**h**): rosette leaf diameter of different *Arabidopsis thaliana*.

**Figure 7 ijms-24-11115-f007:**
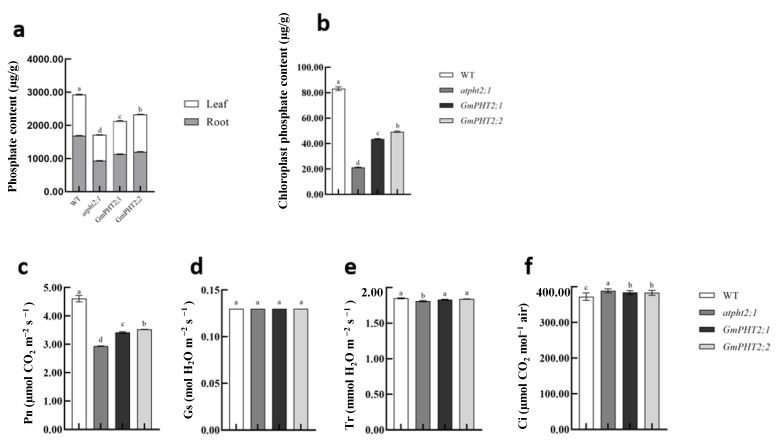
Changes in phosphorus content and photosynthetic indicators in different types of *Arabidopsis thaliana*. (**a**) Phosphorus content of leaves and roots of *Arabidopsis thaliana*; (**b**) chloroplast phosphorus content; (**c**) Pn, net photosynthesis rate; (**d**) Gs, stomatal conductivity; (**e**) Tr, transpiration rate; (**f**) Ci, intercellular CO_2_ concentration. Different lowercase letters indicate significant differences at *p* < 0.05.

## Data Availability

The study’s original contributions are included in the article/Appendix A; additional inquiries should be addressed to the corresponding authors.

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
