# Peer review of "Effects of Soybean Phosphate Transporter Gene GmPHT2 on Pi Transport and Plant Growth under Limited Pi Supply Condition"

_ijms, 2023, doi:10.3390/ijms241311115_

Round 1
Reviewer 1 Report
The manuscript describes experimentations to systematically explore the role of phosphate transporter genes (GmPHT2) on Pi transport and its impact on the plant growth under Pi starvation condition. The discovery expanded our knowledge of mechanisms under phosphate pick up of plants, which may benefit the development of plant quality and yield. The Experiments were performed at a high technical standards and results were clearly addressed. Overall, the manuscript is well-organized and well-written.
Minor comments:
Line 13: RT-PCR cannot discover genes. Authors used this tool for relative gene expression. Genes were identified by bioinformatics. Please rephrase this sentence.
Figure 1: In this reviewer’s opinion, it is better to include the species names on the protein phylogenetic tree.
Results 2.2.: Here is a big jump. Why was yeast assay performed? It was clear in the discussion sectionbut not in the results section.
Figure 4: In figure 4a, Pam2 mutant without any plasmids is missing. In figure 4b, WT is missing. Please keep consistent for pYES2 or empty vector in two panels.
Line 150: Please indicate the meanings of Gal (galactose) and Glu (glucose) in the figure legend.
Line 256: How about the amino acid identify? AA identity is more informative than nucleotide identity when comparing the protein functions.
Line 234: If they are membrane-associated transporters, this reviewer recommended to include the figure of predicted protein structures of GmPHT2;1 and GmPHT2;2, and show where is the transmembrane helix part and where is the domain.
Methods 4.4: The list of primers is missing. Please add the information of gDNA removal method, RNA negative control and water blank control.
Author Response
Comments and Suggestions for Authors
The manuscript describes experimentations to systematically explore the role of phosphate transporter genes (GmPHT2) on Pi transport and its impact on the plant growth under Pi starvation condition. The discovery expanded our knowledge of mechanisms under phosphate pick up of plants, which may benefit the development of plant quality and yield. The Experiments were performed at a high technical standards and results were clearly addressed. Overall, the manuscript is well-organized and well-written.
Minor comments:
Line 13: RT-PCR cannot discover genes. Authors used this tool for relative gene expression. Genes were identified by bioinformatics. Please rephrase this sentence.
Response: Thank you for your kindly suggestion. We have modified it in the manuscript. Please see L13-14.
Figure 1: In this reviewer’s opinion, it is better to include the species names on the protein phylogenetic tree.
Response: Response: Thank you for your kindly suggestion. In Figure 1,Because of the large number of species, in order to express it more clearly so we have added legends to correspond the names of the species to the names of the protein phylogenetic tree.
Results 2.2.: Here is a big jump. Why was yeast assay performed? It was clear in the discussion section but not in the results section.
Response: Thank you for your kindly suggestion. PAM2, a yeast Pi transport mutant deficient in the high-affinity Pi transporter genes PHO84 and PHO89, and facilitate Pi transport across membranes in yeast. Thus, we hypothesized that the GmPHT2 proteins found in soybean might also have Pi transport functions. To test this hypothesis, so we performed yeast assay, The relevant description is in 2.4 of the results.
Figure 4: In figure 4a, Pam2 mutant without any plasmids is missing. In figure 4b, WT is missing. Please keep consistent for pYES2 or empty vector in two panels.
Response: We appreciate the reviewer’s kind suggestions. The Pam2 mutant is a relatively poor transporter of phosphorus under low phosphorus conditions and is essentially unable to grow, so in Figure 4a, the Pam2 mutant without any plasmid is missing. In figure 4b, WT is missing. Thank you for your valuable comments, if WT could be found then this experiment would be perfect. However, our aim was to verify that GmPHT2;1, GmPHT2;2 have complementary yeast inorganic phosphorus transport capacity. As shown by the experimental results, the colour of the yeast cultures transformed with pYES2-GmPHT2;1 and pYES2-GmPHT2;2 was orange-yellow. This indicates that GmPHT2;1 and GmPHT2;2 have the phosphorus transport capacity of the complementary yeast inorganic phosphorus transport mutant PAM2. This is consistent with the results for yeast solid media. In subsequent experiments, we will look for WT to make the results more rigorous. Also, about both panels adjustments were made to pYES2 or the empty carrier to keep it consistent. Please see figure 4
Line 150: Please indicate the meanings of Gal (galactose) and Glu (glucose) in the figure legend.
Response: Thank you for your kindly suggestion. We have indicated the meanings of Gal (galactose) and Glu (glucose) in the figure legend.
Line 256: How about the amino acid identify? AA identity is more informative than nucleotide identity when comparing the protein functions.
Response: Thank you for your kindly suggestion. Based on your suggestion, we have performed both nucleotide and amino acid comparisons and added a new description to the manuscript. Please see L287.
Line 234: If they are membrane-associated transporters, this reviewer recommended to include the figure of predicted protein structures of GmPHT2;1 and GmPHT2;2, and show where is the transmembrane helix part and where is the domain.
Response: Thank you for your kindly suggestion. GmPHT is a membrane-associated transporter and the protein structures of GmPHT2;1 and GmPHT2;2 were predicted and the predictions were added to Figure S1.
Methods 4.4: The list of primers is missing. Please add the information of gDNA removal method, RNA negative control and water blank control.
Response: Thank you for your kindly suggestion. We have added information on all the primers in the manuscript in Table S4, and also in Material Methods 4.3 and 4.4 add the information of gDNA removal method, RNA negative control and water blank control.

Reviewer 2 Report
Wu, Yang and co-workers analyze the Soybean Phosphate Transporter Gene GmPHT2 and evaluate the effect on Pi Transport. They conclude that the two PHT2 analyzed members of soybean are not redundant and that GmPHT2;1 function is not limited to the response to low phosphorus but also to other biological processes s. On the other hand, GmPHT2;2 functions primarily in phosphorus transport.
In general the methodology is sound and accurate, with all the due controls performed according to the usual standards.
What is missing is the quantitative determination of phosphorus, which could be achieved by some of the usual determination methods such as CE or ICP-MS. The quantitative determination in isolated chloroplasts, for instance, could contribute substantially to the evidence already supported.
Arabidopsis thaliana must be in italics, always
use consistency in the abstract: low-phosphate stress / low phosphorus stress
line 14 transport vehicles????
The manuscript lacks a conclusion after materials and methods

Some sentences are completely incomprehensible and a thorough revision by an English speaking person or an editing company is required. See for instance lines 138, 140, 212, 297-298.
Author Response
Reviewer 2:
comments and Suggestions for Authors
Wu, Yang and co-workers analyze the Soybean Phosphate Transporter Gene GmPHT2 and evaluate the effect on Pi Transport. They conclude that the two PHT2 analyzed members of soybean are not redundant and that GmPHT2;1 function is not limited to the response to low phosphorus but also to other biological processes s. On the other hand, GmPHT2;2 functions primarily in phosphorus transport.
In general the methodology is sound and accurate, with all the due controls performed according to the usual standards.
What is missing is the quantitative determination of phosphorus, which could be achieved by some of the usual determination methods such as CE or ICP-MS. The quantitative determination in isolated chloroplasts, for instance, could contribute substantially to the evidence already supported.
Arabidopsis thaliana must be in italics, always
Response: Thank you very much for your useful comments. We have modified it, Arabidopsis thaliana used in italics, at the same time, we checked this kind of problem in the manuscript and corrected it.
use consistency in the abstract: low-phosphate stress / low phosphorus stress
Response: We appreciate the reviewer’s kind suggestions. We have use low-phosphate stress in the abstract, please see L14 and L20.
line 14 transport vehicles????
Response: Thank you for your kindly suggestion. We have changed transport vehicles to transport systems, please see L44.
The manuscript lacks a conclusion after materials and methods
Response: We appreciate the reviewer’s kind suggestions. We have added conclusion after materials and methods. please see L471-478.
Comments on the Quality of English Language
Some sentences are completely incomprehensible and a thorough revision by an English speaking person or an editing company is required. See for instance lines 138, 140, 212, 297-298.
Response: Thank you for your kindly suggestion. We have redescribed these sentences to make the meaning of their expression clearer, in addition, before this revision, the language of the manuscript has been edited by native English experts.

Round 2
Reviewer 2 Report
The authors ignored my comment that remains the same: What is missing is the quantitative determination of phosphorus, which could be achieved by some of the usual determination methods such as CE or ICP-MS. The quantitative determination in isolated chloroplasts, for instance, could contribute substantially to the evidence already supported.
The authors ignored my comment that remains the same: What is missing is the quantitative determination of phosphorus, which could be achieved by some of the usual determination methods such as CE or ICP-MS. The quantitative determination in isolated chloroplasts, for instance, could contribute substantially to the evidence already supported.
Author Response
The authors ignored my comment that remains the same: What is missing is the quantitative determination of phosphorus, which could be achieved by some of the usual determination methods such as CE or ICP-MS. The quantitative determination in isolated chloroplasts, for instance, could contribute substantially to the evidence already supported.
Response: First of all, I am very, very sorry for not answering your comment and thank you again for your valuable comments. Regarding the determination of phosphorus content in the manuscript, we used the ascorbate-molybdate-antimony method, referring to the experimental method of John, 1970 to determine the phosphorus content in transgenic Arabidopsis thaliana. This is described in the manuscript Materials and Methods 4.4 L461.The results in Fig. 7 show that complementation and overexpressed plants lower phosphorus content than wild-type plants but significantly higher than mutant plants (Figure 7a). The phosphorus content of the chloroplasts, above-ground and below-ground parts of the plants in the back-complemented GmPHT2;2 plants were considerably higher than in the back-complemented GmPHT2;1 plant (Figure 7a, 7b). and combined with changes in photosynthetic data suggest that the GmPHT2;1 and GmPHT2;2 genes restore the photosynthetic capacity of Arabidopsis mutant plants to some extent by increasing the phosphorus content of chloroplasts in the Arabidopsis mutant.

Round 3
Reviewer 2 Report
The methods used is not the best but is is sufficient to have a further proof of evidence. The manuscript is suitable for publication
the English can be improved